# Evaluation of the Uncertainty of Surface Temperature Measurements in Photovoltaic Modules in Outdoor Operation

**DOI:** 10.3390/s22155685

**Published:** 2022-07-29

**Authors:** Carmen García-López, Germán Álvarez-Tey

**Affiliations:** Departamento de Ingeniería, Eléctrica, Universidad de Cádiz, Avenida de la Universidad de Cádiz, 10, 11519 Cádiz, Spain; german.alvarez@uca.es

**Keywords:** PV temperature measurement, degradation, uncertainty, accurate and precision analysis

## Abstract

Faults in photovoltaic modules in operation can lead to power losses. By determining the module surface temperature, hot spots that can potentially cause this power loss can be detected. Temperature measurement by radiation allows a complete, reliable, and fast qualitative determination of hot spots on PV modules in outdoor operation. However, to obtain quantitative values, it is necessary to consider multiple factors: emissivity, reflected radiation, wind speed, intensity, shading, etc. Temperature quantitative measurement evaluation by contact is more studied, although by this technique it is impossible to examine the temperature of the entire module to detect hot spots because it is a point measurement and due to shading caused by the measurement probe on the surface. In this work, a method of temperature measurement by radiation is described, evaluating the uncertainty components, and a comparison is made with temperature measurement by contact on the module rear side points where module heating has been detected, also evaluating the uncertainty components. This comparison of both methods and uncertainty determination allows establishing a methodology in quantitative temperature measurement by radiation in photovoltaic modules in outdoor operation.

## 1. Introduction

Photovoltaic (PV) plants can be affected by various causes [1]. These malfunctions are associated with an increase in temperature in different areas of the PV module [2]; thus, a study of module temperature allows the detection of areas in which a failure occurs [3]. A complete review of correlations that express the efficiency of a PV system as a function of PV cell temperature and meteorological variables such as ambient temperature, local wind speed, and solar irradiance, with material and system-dependent properties as parameters, is studied in [4].

Infrared thermography (IR) is a non-contact temperature evaluation method that allows obtaining PV module thermal behavior. Several works in which qualitative thermographic inspections are carried out to detect defects in PV modules that can cause power loss [5,6,7,8], although not all faults can be detected by thermography [9]. However, quantitative analysis and a measurement uncertainty components estimation are necessary to establish acceptance and rejection criteria for IR measurement in PV systems [10]. Non-contact temperature quantitative measurement is applied by some authors in the study of PV plants [11], but it is also applied to other fields. In [12], the authors apply this method to body temperature measurement from a clinical and metrological point of view, intending to improve the accuracy of body temperature measurements and estimating body temperature measurement uncertainty in the field, all to propose a screening decision rule for prevention of the spread of COVID-19. Another field of application in building is studied in [13], where authors present a method to quantify reflected flux using an infrared mirror, which allows estimating the temperature of large surfaces using IR thermography in almost ambient conditions with improved accuracy.

Uncertainty evaluation is well studied in [14], where a detailed metrological analysis is carried out to evaluate various causes of error (due to measurement, instrument, environment, and operator) and estimate measurement uncertainty. General rules applicable to a wide spectrum of measurements are established. A study of IR thermography fundamentals, algorithms used by IR cameras for measurement expression, measurement errors analysis, and measurement uncertainty for both correlated and uncorrelated input variables are studied in detail in [15].

In PV plants in outdoor operation, the application of thermographic measurements requires errors study that does not occur in indoor measurements. When performing exterior thermographs, there is a reflection of infrared rays on glass that have to be taken into account. In [16], this effect is studied, and a correction method for these errors is proposed and experimentally validated. A method is also presented to estimate an equivalent sky temperature that can be used in the procedures for outdoor thermographic measurements correction. A procedure for the correct configuration of an outdoor thermographic measurement system is presented in [17], where proper positioning of thermographic equipment is studied to minimize reflections from the sun and sky. For this, ideal inspection height and minimum height according to the layout of the PV installation are studied. In [18], the authors analyze the error sources in outdoor PV systems evaluation. The objective of this work is to determine and quantify error sources associated with outdoor tests and propose procedures and techniques to minimize them. The technical specification IEC 62446-3 [19] defines and establishes outdoor IR thermography of PV plants in operation. This IEC 62446-3 specification considers the requirements for measurement equipment, environmental conditions, inspection procedure, inspection report, personnel qualification, and a matrix for thermal anomalies as a guide for inspection. This specification aims to harmonize and standardize inspection methodology to facilitate comparing results.

Temperature measurements by contact in photovoltaic systems have to aim to accurately compare these measurements with temperature measurements by non-contact methods [20]. Such comparisons are made in other fields, such as in the case of electrical power system transmission lines [21], in which temperature measurements by contact and non-contact are performed for several months under different climatic and load conditions, with temperature measurement by IR in the power system transmission line conductor being in a range of less than 4 °C difference concerning contact temperature measurement methods for 88% of the samples and less than 6 °C for 99% of the samples.

The objective of this study is to establish a methodology for obtaining quantitative temperature measurements on photovoltaic modules in outdoor operation using infrared thermography, evaluating the measurement uncertainty components. This measurement is compared with measurements made with contact sensors on the module’s rear side and their uncertainty components. The study has been carried out on monocrystalline and polycrystalline modules.

## 2. Materials and Methods

The tests were carried out with a monocrystalline module (LUXOR LX195M/125-72 +) and a polycrystalline module (AXITEC AC-250 P/60S) in which hot spots were caused. The modules show no significant defects. Some cells of the photovoltaic modules have been covered with a material of known emissivity so that this module area is heated and temperature can be quantitatively determined by infrared thermography. The material used to cover cells was polyethylene adhesive tape. To carry out the IR inspection, it is necessary to know this material’s emissivity. Its emissivity was obtained experimentally according to the method established in [22]. An emissivity value of 0.92 was obtained.

In each test performed, each module was exposed to external environmental conditions and connected to an electronic load that provides a thermal behavior close to the maximum power in the corresponding IV curve. Once the temperature was stabilized, it was measured by radiation on the front side. In those areas where hot spots could be detected, the temperature was measured by contact thermocouples on the module’s rear side. The temperature measured in both measurement by contact and measurement by radiation corresponds to the cell’s central zone. The measurements by contact and by radiation were simultaneously performed in these cells.

The cells covered were those shown in Figure 1 and were identified as E3, C4, and B6 in both module types.

### 2.1. Measurement by Contact

To evaluate measurement by contact, an exposed junction type K thermocouple was selected. This measurement cannot be performed on the PV module front side, as it would produce a shadow on the module, so the measurement solution on the module rear side was chosen. The temperature recording was performed at 1-min intervals for 75 min. Thermocouples were fixed to the PV module surface with aluminum tape, which has high thermal conductivity (Figure 2).

Data acquisition was performed with Fluke Hydra 2625A multipoint recorder equipment.

### 2.2. Measurement by IR

To evaluate the PV module front side temperature, a FLIR ThermaCAM S60 was used. This camera has a spectral range from 7.5 to 13 μm and a resolution of 320 × 240 pixels. By this, the IR camera was placed at an angle α slightly greater than 0° to avoid self-reflection and less than 40° from a perpendicular line so that emissivity could be considered constant (Figure 3). According to IEC 62446-3 [19], thermographic inspection was carried out under an irradiance higher than 700 W/m^2^, low wind speeds, and an absence of cloudiness. A weather station was used to record environmental data during the test. Furthermore, to set an accurate camera configuration to reduce measurement errors, the emissivity and the apparent reflected temperature were evaluated as shown in [3,22].

Thermographic equipment was also configured with a recording interval of 1 min for 75 min. The registered thermographic images were analyzed to obtain temperature values. Figure 4 shows one of the samples where hot spots were detected in the monocrystalline module. These points were analyzed in each of the recorded images.

All the thermal and electrical equipment used in the tests have calibration certificates that provide metrological traceability with higher-level standards.

### 2.3. Evaluation of Uncertainty Components

Measurement uncertainty is a parameter associated with a measurement result, which characterizes values dispersion that could reasonably be attributed to the measurand.

Uncertainty is generally composed of several components. Some of them are evaluated from statistical distributions of measurement results. They can be characterized by their experimental standard deviations (type A uncertainties). Others arise from systematic effects, such as the process itself or measuring equipment used, which can also be evaluated assuming probability distributions based on experience or other information (type B uncertainties).

Measurement results are the best estimate of the measurand value, accompanied by a statement of all the uncertainty components contributing to its dispersion.

In measurements performed, the values of the measurand are the temperatures obtained in measurement by the contact method (*T_c_*) and measurement by a radiation method (*T_r_*). The uncertainty is obtained from evaluating dispersion sources involved in the measurement process.

The main uncertainty components in this type of measurement are discussed below. Although there are other uncertainty sources, their values are negligible compared to the most important components [14,15].

#### 2.3.1. Uncertainty in Measurement by Contact

In the temperature measurement system by contact used in these tests, the thermocouples adhered to the module rear side do not reach thermal stability, so the measurement has to be considered at each instant of time. The value of the measurand is considered as a value at each instant of time and in each position of readings provided by the recording equipment of each sensor. And each temporal and spatial measurement is associated with an uncertainty that comes from the equipment used and the process itself. The components of uncertainty that have been taken into account are expressed below:

Calibration uncertainty of data logger measuring instrument (*T_cal_*): The uncertainty declared in the calibration certificate. In instrument calibration, internal cold junction compensation in measurement is included as an uncertainty component. This uncertainty can be considered a normal distribution. The uncertainty is given with a confidence level of 95%, with a coverage factor k = 2, so divisor 2 is necessary to obtain the standard uncertainty.Data recording equipment uncertainty due to multiple effects (*T_d_*): Drift, linearity, stability, load, supply voltage variation, and calibration error. To consider these effects, data of the manufacturer’s one-year accuracy specifications will be taken, considering that instrument reading will be between the upper and lower limits given by specifications with the same probability. Therefore, a rectangular distribution will be considered. To take into account the effects of ambient temperature on measuring equipment resulting from temperature coefficient temperature variations occurring during use, ambient temperature during the test shall be recorded. If this temperature is outside the operating specification range of 18 °C to 28 °C, take the specifications from 0 °C to 60 °C to account for this deviation.Cold junction compensation uncertainty (*T_j_*). The data logger measuring instrument employs an internal cold junction compensation. The uncertainty associated with the measurement using this cold junction compensation shall be considered, and the deviation of the calibration certificate shall be taken as a rectangular distribution.Resolution uncertainty (*T_resol_*): This is the uncertainty associated with digital scopes. The actual reading is equiprobable between the lower and upper limits, so a rectangular distribution is assumed.Uncertainty of measurement thermocouples (*T_T_*): The test thermocouples are purchased from different suppliers with different calibration uncertainty ranges. The thermocouples are purchased in 600 m coils calibrated at both ends. They are then cut to the desired length. All thermocouples used in tests have class 1 tolerance according to IEC 60584-1:2013 [23]. This tolerance will be taken as a rectangular distribution.Uncertainty due to measurement system delay (*T_S_*): In a stabilized system, this component is negligible. However, in non-stabilized systems, maximum deviation due to delay introduced by the measurement system can be considered to be the difference between the next and current measurements, taken as a rectangular distribution.Uncertainty due to surface measurement (*T_i_*): When determining surface temperature, it has to be taken into account that it is not an isothermal medium being considered but a thermocouple that sticks to the surface. Surface measuring thermocouples, according to Nicholas, have an error of between 5% and 10% [24]. This value will be taken as the maximum value of rectangular distribution.Uncertainty due to thermocouple inhomogeneity (*T_h_*): This component is evaluated as the difference between the deviation in calibration at the beginning and end of the thermocouple roll, obtained from the calibration certificate and considered as a rectangular distribution.

Table 1 summarises uncertainty components to be applied in measurement. A spreadsheet has been prepared for the application of these components. The combined standard uncertainty is shown in Equation (1), and the expanded uncertainty with coverage factor k = 2 in Equation (2).
(1)$$u\left(T_{c}\right)=\sqrt{u{\left(T_{cal}\right)}^{2}+u{\left(T_{d}\right)}^{2}+u{\left(T_{j}\right)}^{2}+u{\left(T_{resol}\right)}^{2}+u{\left(T_{T}\right)}^{2}+u{\left(T_{s}\right)}^{2}+u{\left(T_{i}\right)}^{2}+u{\left(T_{h}\right)}^{2}}$$
(2)$$U\left(T_{c}\right)=2\;u\left(T_{c}\right)$$

#### 2.3.2. Uncertainty in Measurement by IR

Temperature quantitative measurement by IR thermography is still a wide field of research. Many factors are involved in converting thermal radiation obtained at the camera detector to a temperature. Calibration procedures for thermographic cameras are based on measuring black bodies at different temperatures so that external factors to the camera itself, such as emissivity, atmosphere, and apparent reflected temperature, are negligible [25]. However, these factors are critical in converting radiation to temperature in a thermal image in a real test that does not correspond to the image of a blackbody.

The study of uncertainty components in thermographic camera measurements is a field of research in radiometric metrology. The BIPM non-contact thermometry working group CCT-WG-NCTh [26] has, among its objectives, developed a suitable uncertainty balance for radiation thermometry, focusing on the calibration of single-point detectors. A field of interest is the study of uncertainty in array detectors, such as thermographic cameras.

In an indoor laboratory environment, uncertainty components in thermographic measurement depend exclusively on measuring equipment: accuracy, thermal resolution, drift between calibrations, non-uniformity, source size, and distance effect. Under these conditions, the uncertainty components associated with measurement can be determined.

However, in real outdoor measurements, as in this work, it is especially important to consider the uncertainty components associated with the process: emissivity calculation, atmospheric effect, and reflection of the environment [15].

Of all factors contributing to overall measurement uncertainty, the uncertainty due to determining the material’s emissivity must be highlighted. Although there are tabulated emissivity values for different materials, this parameter depends on wavelength, temperature, material, surface finish, and direction of observation. In very fast thermal processes, it is also time-dependent. Therefore, determining an emissivity value for the object in question has to be associated with an uncertainty covering all these factors.

Errors in measurement by radiation thermometry are mainly due to factors described below:
Errors due to the measurement method. In real conditions, the main sources of error due to measurement methods are emissivity evaluation, the influence of the radiation emitted by the environment, and transmission through the atmosphere, Figure 5. The lens of the thermal imaging camera detects radiation energy that is proportional to the energy emitted by the object (W_obj_), energy reflected by the object itself (W_refl_), and energy transmitted by the atmosphere (W_atm_). If there are different objects with different emissivity in an image, the effect of emissivity on error evaluation is more noticeable. In thermographic image analysis software, the emissivity of different areas can be specified separately, and post-data acquisition analysis can be performed, so if images are taken with emissivity equal to 1, this error is reduced during the image acquisition process. Subsequently, different areas of the image are evaluated with emissivity values obtained in each case and their corresponding uncertainties.

Eliminating the influence of an incorrect evaluation of emissivity can be complicated. In the stabilized thermal processes, it is possible to reduce the influence of incorrect emissivity assessment by painting the surface of objects black or covering them with a material with emissivity close to unity. So that when temperature stabilizes, measurement on a material surface with an emissivity value close to unity is obtained in a thermal imaging camera. Since the evaluation of emissivity is one of the greatest sources of error in the measurement carried out with a thermographic camera, it requires a detailed study to determine the error limits and establish an approximation using a rectangular distribution. For this purpose, an emissivity estimate has been made on the material of which the module surface is covered.

Another source of error due to the measurement method is the influence of radiation emitted or reflected by bodies around the object to be measured. This radiation is greater the lower the emissivity and reflectivity of the object. Its effect is more significant at object temperatures close to ambient temperature and which have a suitable angle of incidence on the object to become part of outgoing radiation into the IR camera. If measurements are made outdoors, it is necessary to consider components due to the sky, which will consider atmospheric transmission.

The apparent reflected temperature, coming from the rest of the bodies around the test, is evaluated around the measurement point before the test by obtaining temperature without any compensation, i.e., emissivity at 1 and distance at 0 over a Lambert radiator.

Errors due to instrument calibration: Equipment used in measurements has a calibration certificate issued with traceability by a laboratory with accreditation recognized by International Laboratory Accreditation Cooperation (ILAC). This accreditation ensures that the calibration method minimizes errors that can be made in poor detector calibration. A calibration certificate provides an uncertainty component of measurement referred to as a standard that has to be considered in measurements. In addition, to account for drift, hysteresis, and other sources of error, the accuracy specifications of thermal imaging cameras shall be considered. When making measurements, the numerical resolution of results is a source of the truncation error that shall be considered.Errors due to instrument electronics: These systematic errors result from factors due to instrument electronics, such as detector noise, instability or fluctuations in amplifiers gain and/or other electronics, limitation of detector bandwidth, and limitation and non-linearity of A/D converters. This error is below ±1% at ambient temperatures from −15 °C to 40 °C.Spatial resolution: Spatial resolution is a characteristic parameter of each piece of equipment we obtain regarding its specifications. It determines the best resolution that can be obtained in the image or area represented by each digital image pixel. The detector of the thermal imaging camera used has a resolution of 320 × 240 pixels. From specifications it can be seen that for test distance, Figure 6, D = 5.0 m field of view parameters are HFOV = 2.13 m, VFOV = 1.59 m and IFOV = 6.64 mm.

To balance the uncertainty components in measurement by radiation, these sources of error shall be considered. An estimate shall be made based on available data, the metrological experience of the Laboratory, and experimental data. It has to be noted that the main sources of error are the determination of the material emissivity and the camera spatial resolution.

Main uncertainty components in temperature measurement by radiation are due, on the one hand, to measuring equipment itself and, on the other hand, to errors in the measuring method:

Calibration uncertainty of measuring instrument thermal imaging camera (*T_cal_*): This is the uncertainty stated in the calibration certificate. This uncertainty can be considered a normal distribution. Uncertainty is given at a 95% confidence level, with coverage factor k = 2, so divisor 2 is necessary to obtain standard uncertainty.Thermal imaging camera equipment uncertainty due to multiple effects (*T_d_*): Drift, linearity, stability, load, supply voltage variation, and calibration error. To account for these effects, data from the manufacturer’s one-year accuracy specifications shall be taken, understanding that instrument reading shall lie between upper and lower limits given by specifications with the same probability. Therefore, a rectangular distribution shall be considered. To account for the effects of ambient temperature on measuring equipment resulting from temperature coefficient and temperature variations experienced during use, ambient temperature shall be recorded during the test. Normal operating range is −15 °C to 40 °C (±2 °C or 2% of reading).Digital display resolution uncertainty (*T_resol_*): This is the uncertainty associated with digital displays. The actual reading is equiprobable between lower and upper limits, and thus a rectangular distribution is assumed.Uncertainty due to detector electronics (*T_e_*): This component under normal operating conditions is not more than 1%.Spatial resolution (IFOV) (*T_IFOV_*): The number of points to be obtained in an image corresponding to the area to be evaluated is the ratio between the total area to be considered in the module surface and the maximum area to be obtained in each measurement pixel. To evaluate this component, minimum and maximum values are determined in this image fraction and are considered as error limits in a rectangular distribution.Uncertainty due to the determination of the object’s emissivity (*T_ε_*): It is the most important uncertainty component. The emissivity is determined by comparison on a specimen of the same material as the object to be measured. It includes uncertainty associated with measurement by radiation and by contact and comparison. Error limit with rectangular distribution is the maximum percentage variation in tests at different temperatures. The temperature variation corresponding to the maximum emissivity variation obtained experimentally is considered. This component comprises the precision of the IR camera used for measurements, together with the contact temperature measurement (thermocouple, acquisition equipment).Uncertainty due to apparent reflected temperature (*T_rf_*): This component can be minimized by adjusting the corresponding IR camera parameter before the test. The parameter is set to a value determined by the initial evaluation of the IR camera, capturing surrounding radiation on a Lambert radiator without compensation, i.e., with emissivity parameter set to 1 and distance parameter set to 0. This component is considered the upper and lower limits in rectangular distribution.Uncertainty due to atmospheric transmission (*T_at_*): During the test, it is necessary to record the ambient conditions under which the equipment performs measurements. The value of ambient temperature and relative humidity shall be set in the IR camera parameters to compensate for transmission through the atmosphere. In the test, the IR camera shall be placed at a distance of 5 m from the module.

Table 2 summarises uncertainty components to be applied in measurement by radiation. The combined standard uncertainty obtained for this measurement is calculated from Equation (3), and the expanded uncertainty with coverage factor k = 2 from Equation (4).
(3)$$u\left(T_{r}\right)=\sqrt{u{\left(T_{cal}\right)}^{2}+u{\left(T_{d}\right)}^{2}+u{\left(T_{resol}\right)}^{2}+u{\left(T_{e}\right)}^{2}+u{\left(T_{IFOV}\right)}^{2}+u{\left(T_{\varepsilon }\right)}^{2}+u{\left(T_{rf}\right)}^{2}+u{\left(T_{at}\right)}^{2}}$$
(4)$$U\left(T_{r}\right)=2u\left(T_{r}\right)$$

## 3. Results

The temperature measurements were taken according to the methodology described, using thermography on the module front side (*T_r_*) and a contact sensor on the module rear side (*T_c_*) for 75 min.

Figure 7 shows results obtained for *T_r_* and *T_c_*, as well as the uncertainty, *U*(*T_c_*) and *U*(*T_r_*), of measurements made in each of the cells where hot spots were detected in the monocrystalline module.

A summary of the values obtained in the contact temperature measurements on the monocrystalline module is shown in Table 3, where *T_c-max_* and *T_c-min_* correspond to maximum and minimum temperature values, respectively, obtained during data recording. And in Table 4, values obtained in the thermographic temperature measurements on the same module are shown, where *T_r-max_* and *T_r-min_* have the same meaning as contact temperature measurement.

The measurement has been carried out, following the same procedure, on the polycrystalline module. Figure 8 shows the results obtained, and a summary of the results is shown in Table 5 for the measurement by contact and Table 6 for the measurement by radiation. *T_c_**, T_r_*, *U*(*T_c_*), *U*(*T_r_*), *T_c-max_*, *T_c-min_*, *T_r-max_*, and *T_r-min_* have the same meaning that in the monocrystalline module.

The uncertainties obtained in the temperature measurement by radiation are larger than those obtained in the temperature measurement by contact. There are many factors involved in determining the uncertainty in measurement by radiation. However, the values obtained are consistent with the results obtained for the temperature measurement by contact.

In each case and for each record, the difference between the temperature measurement by contact taken on the module rear side (*T_c_*) and measurement taken by thermography (*T_r_*) has been calculated, Equation (5).
(5)$$\Delta T_{i}=T_{ci}-T_{ri}\;\;1\leq i\leq 75$$

And the uncertainty for this measure is,
(6)$$U\left(\Delta T\right)=2\;\sqrt{{\left(\frac{U\left(T_{c}\right)}{2}\right)}^{2}+{\left(\frac{U\left(T_{r}\right)}{2}\right)}^{2}}$$

Figure 9 shows the results for the monocrystalline module. Graphs (a), (c), and (e) show the temperature values recorded in the module (*T_c_* and *T_r_*) and temperature difference (Δ*T*) during 75 min of recording. In plots (b), (d), and (f), the approximation of the values to a normal distribution of Δ*T* of the measurement at each hot spot has been plotted.

Figure 10 shows the results for the polycrystalline module. Graphs (a), (c), and (e) show the temperature values recorded in modules (*T_c_* and *T_r_*) and temperature difference (Δ*T*) during the 75 min of recording. In plots (b), (d), and (f), an approximation of the data to a normal distribution of Δ*T* has been plotted.

Although the tests performed correspond to different cells, and an average value has been obtained for each of them, Figure 11 shows a comparison of the result of the approximation to a normal probability distribution function of each cell for the two types of tested modules.

Considering all data as a whole gives the histogram and the approximation to the normal distribution function shown in Figure 12.

For the monocrystalline module, the values obtained for Δ*T* fit a normal probability distribution. The temperature obtained in measurement by radiation remains, on average, 8.3 °C below values obtained in measurement by contact, and with a maximum uncertainty, according to Equation (6), of 14 K. It can be concluded that the temperature difference between measurement by contact and measurement by radiation is below 10 °C for 87% of the samples and below 12.8 °C for 99% of them.

In the case of the polycrystalline module, the fit of Δ*T* to normal probability distribution shows that this difference is, on average, 6.4 °C., with a maximum uncertainty, according to Equation (6), of 11.4 K. It is observed in this case that temperature measured by radiation, although still lower than that obtained in measurement by contact, is lower than that obtained in case of the monocrystalline module. The Δ*T* difference is below 8 °C for 89% of the samples and below 10.3 °C for 99% of them.

## 4. Discussion

The methodology described is applied in the same way to both monocrystalline and polycrystalline modules. This method estimates uncertainty in temperature measurements obtained by radiation on the module front side and in measurements obtained on the module rear side by contact. Although this work has analyzed hot spots caused by the covering of the cells, it extends to the existence of hot spots due to multiple causes. Instead of using the emissivity value of the polyethylene tape, the module glass emissivity would be used.

Uncertainty values obtained in temperature measurements by radiation are higher than uncertainty values obtained in temperature measurements by contact.

In all cases, temperature measurements obtained by radiation are lower than those obtained on the module’s rear side. In polycrystalline modules, this difference is smaller than in monocrystalline modules. The module’s front and rear sides are subject to different external conditions, so temperatures are not the same.

The differences between monocrystalline and polycrystalline modules are due to the different materials in the modules and the fact that the two types of modules have different thermal characteristics.

Temperature measurement by radiation makes it possible to qualitatively determine hot spots of a PV module operating outdoors, where faults occur. Using the study carried out in this work, temperature measurement by radiation and an estimate of uncertainty can be quantitatively obtained. The application of this method allows, concerning thermographic inspections in photovoltaic plants, to quantitatively evaluate the temperature in hot spots detected. These measurements, compared with those obtained on the module’s rear side, allow an estimate of the temperature difference that a module with similar characteristics to those studied would have between the front and the rear face.

## Figures and Tables

**Figure 1 sensors-22-05685-f001:**
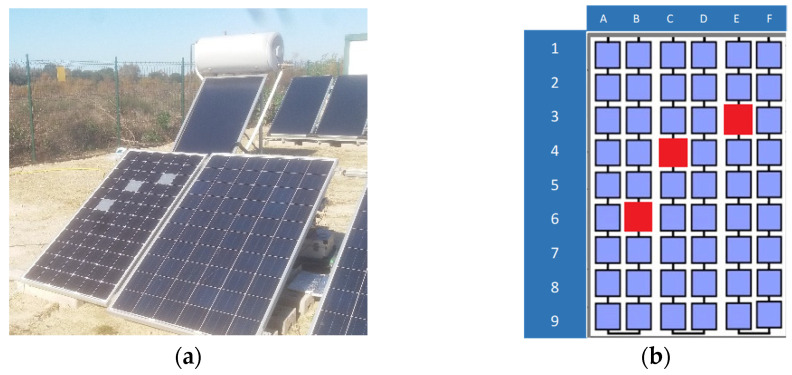
(**a**) Cells covered on the module. (**b**) Numbering of the cells for identification purposes.

**Figure 2 sensors-22-05685-f002:**
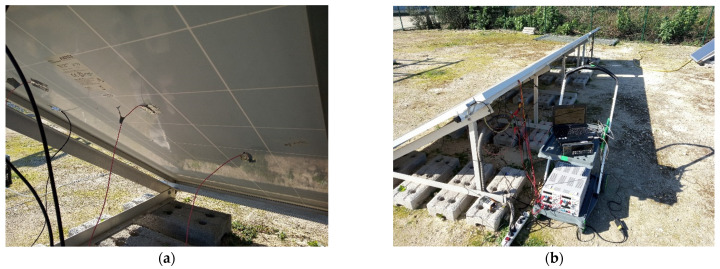
Contact temperature measurement method (**a**) Thermocouple mounting on the rear side of the module; (**b**) Data logging equipment.

**Figure 3 sensors-22-05685-f003:**
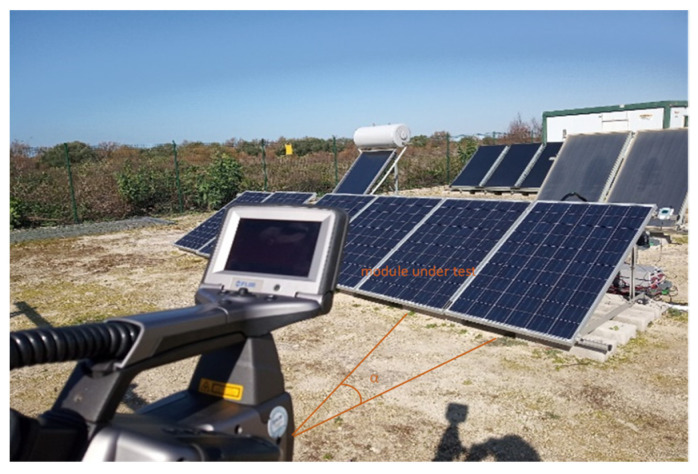
Position of Thermal Camera in front of PV module with α between 0° and 40°.

**Figure 4 sensors-22-05685-f004:**
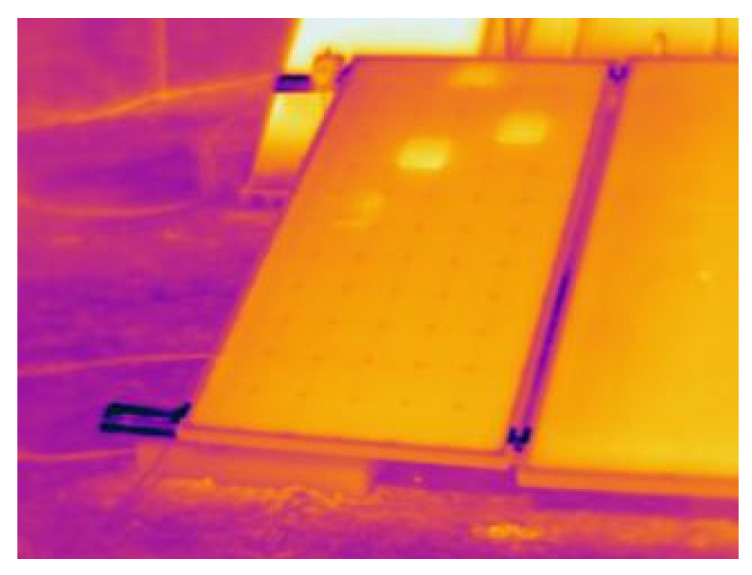
Example of a thermal image obtained in the tests.

**Figure 5 sensors-22-05685-f005:**
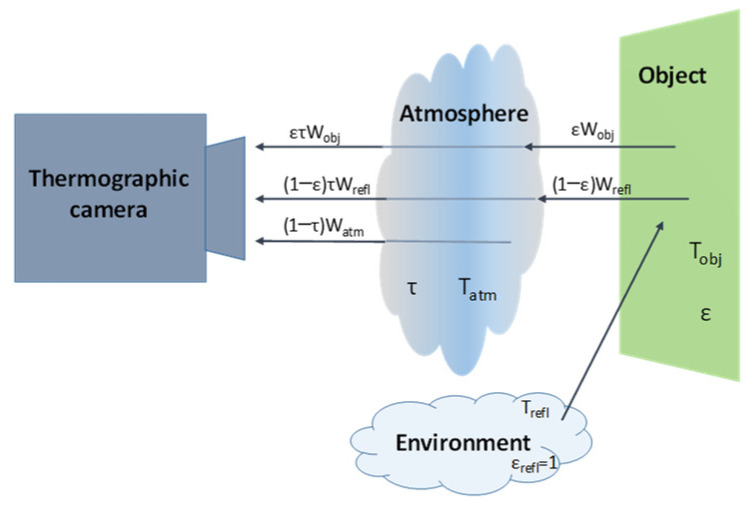
Elements involved in evaluating uncertainty in thermographic measurements: environment, object surface, atmosphere, and camera receiver.

**Figure 6 sensors-22-05685-f006:**
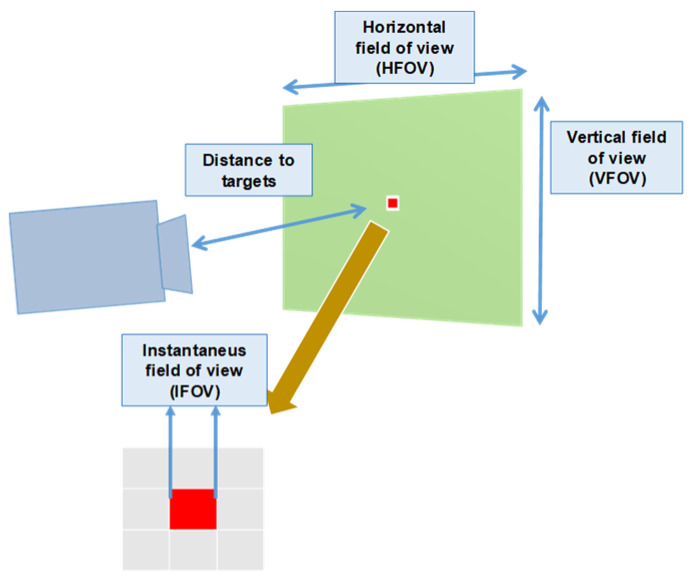
Field of view of the thermal imaging camera. Distance from lens to object determines the spot size represented in each camera pixel (IFOV).

**Figure 7 sensors-22-05685-f007:**
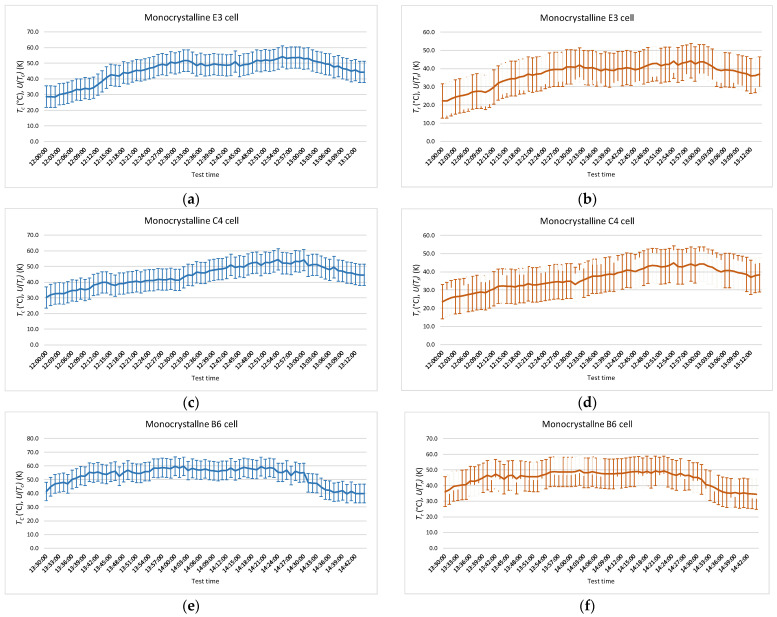
Temperatures obtained during the test in monocrystalline modules. The temperatures measured by radiation and by contact are shown with their uncertainty. (**a**) *T_c_* in E3 cell; (**b**) *T_r_* in E3 cell; (**c**) *T_c_* in C4 cell; (**d**) *T_r_* in C4 cell; (**e**) *T_c_* in B6 cell; (**f**) *T_r_* in B6 cell.

**Figure 8 sensors-22-05685-f008:**
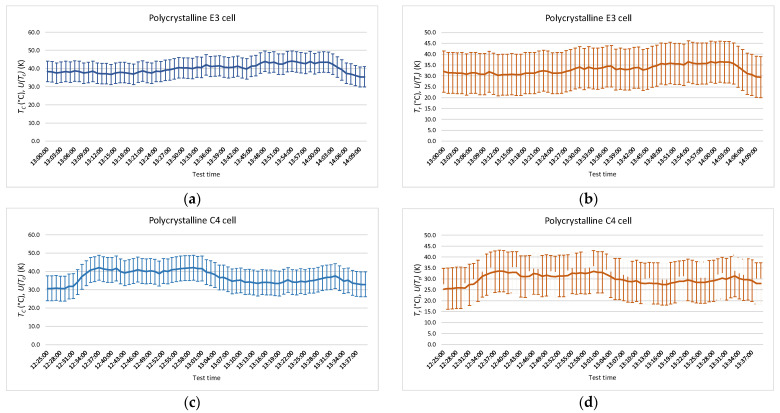
Temperatures obtained during the test in polycrystalline modules. The temperatures measured by radiation and by contact are shown with their uncertainty. (**a**) *T_c_* in E3 cell; (**b**) *T_r_* in E3 cell; (**c**) *T_c_* in C4 cell; (**d**) *T_r_* in C4 cell; (**e**) *T_c_* in B6 cell; (**f**) *T_r_* in B6 cell.

**Figure 9 sensors-22-05685-f009:**
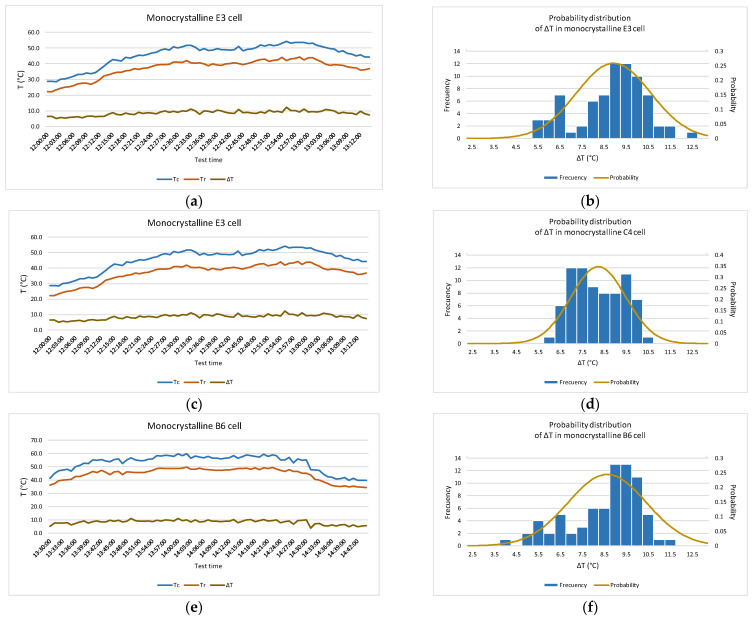
Temperatures obtained during test in a monocrystalline module. Measured temperature by radiation and by contact are shown together with the difference between them (Δ*T*) in the left-wing graph. The histogram and the approximation to a normal probability distribution function of Δ*T* are shown in the right-wing graph. (**a**,**b**) in E3 cell; (**c**,**d**) in C4 cell; (**e**,**f**) in B6 cell.

**Figure 10 sensors-22-05685-f010:**
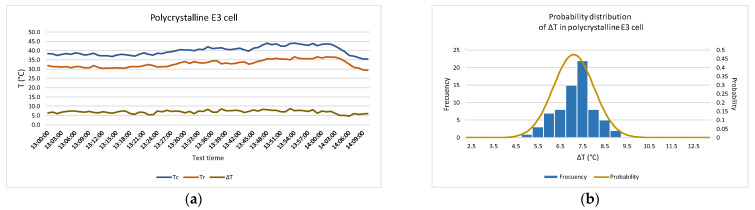
Temperatures obtained during the test in the polycrystalline module. Measured temperature by radiation and by contact are shown together with the difference between them (Δ*T*) in the left-wing graph. The histogram and the approximation to a normal probability distribution function of Δ*T* are shown in the right-wing graph. (**a**,**b**) in E3 cell; (**c**,**d**) in C4 cell; (**e**,**f**) in B6 cell.

**Figure 11 sensors-22-05685-f011:**
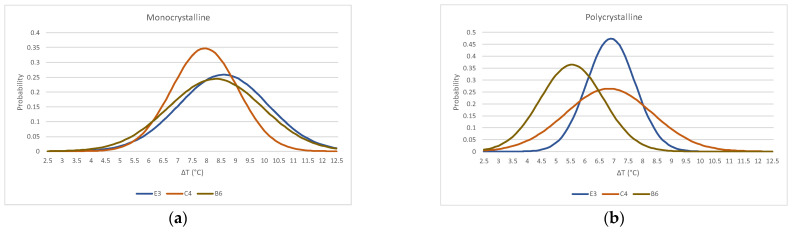
Comparison of cell’s normal probability distribution function (**a**) monocrystalline module (**b**) polycrystalline module.

**Figure 12 sensors-22-05685-f012:**
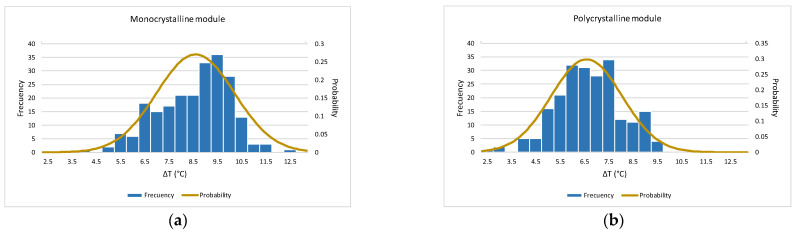
Normal probability distribution function (**a**) monocrystalline module (**b**) polycrystalline module.

**Table 1 sensors-22-05685-t001:** Uncertainty components in measurement by contact.

Uncertainty Sources	Value (°C)	Distribution	Divider	Standard Uncertainty (K = 1) (K)
Data logger calibration	*T_cal_*	normal	2	$u\left(T_{cal}\right)=\frac{T_{cal}}{2}$
Drift, linearity	*T_d_*	rectangular	√12	$u\left(T_{d}\right)=\frac{T_{d}}{\surd 12}$
Compensation cold junction	*T_j_*	rectangular	√12	$u\left(T_{j}\right)=\frac{T_{j}}{\surd 12}$
Resolution	*T_resol_*	rectangular	√12	$u\left(T_{resol}\right)=\frac{T_{resol}}{\surd 12}$
Thermocouple. Maximum deviation Class I	*T_T_*	rectangular	√12	$u\left(T_{T}\right)=\frac{T_{T}}{\surd 12}$
Measurement system delay	$T_{s}=T_{t}-T_{t-1}$	rectangular	√12	$u\left(T_{s}\right)=\frac{T_{s}}{\surd 12}$
Surface measurement	*T_i_*	rectangular	√12	$u\left(T_{i}\right)=\frac{T_{i}}{\surd 12}$
Inhomogeneity	*T_h_*	rectangular	√12	$u\left(T_{h}\right)=\frac{T_{h}}{\surd 12}$

**Table 2 sensors-22-05685-t002:** Uncertainty components in measurement by radiation.

Uncertainty Sources	Value (°C)	Distribution	Divider	Standard Uncertainty (K = 1) (K)
Camera calibration	*T_cal_*	normal	2	$u\left(T_{cal}\right)=\frac{T_{cal}}{2}$
Drift and linearity	*T_d_*	rectangular	√12	$u\left(T_{d}\right)=\frac{T_{d}}{\surd 12}$
Resolution digital display	*T_resol_*	rectangular	√12	$u\left(T_{resol}\right)=\frac{T_{resol}}{\surd 12}$
Detector electronics	*T_e_*	rectangular	√12	$u\left(T_{e}\right)=\frac{T_{d}}{\surd 12}$
Spatial resolution	*T_IOFV_*	rectangular	√12	$u\left(T_{IOFV}\right)=\frac{T_{IOFV}}{\surd 12}$
Emissivity	*T_ε_*	rectangular	√12	$u\left(T_{\varepsilon }\right)=\frac{T_{\varepsilon }}{\surd 12}$
Apparent reflected temperature	*T_rf_*	rectangular	√12	$u\left(T_{rf}\right)=\frac{T_{rf}}{\surd 12}$
Atmospheric transmission	*T_at_*	rectangular	√12	$u\left(T_{at}\right)=\frac{T_{at}}{\surd 12}$

**Table 3 sensors-22-05685-t003:** Temperature values obtained from measurements by contact on the monocrystalline modules.

Cell	*T_c-max_* (°C)	*T_c-min_* (°C)	*U*(*T_c_*) (K)
E3	54.2	28.5	6.8
C4	54.5	30.2	7.5
B6	59.9	39.7	10.3

**Table 4 sensors-22-05685-t004:** Temperature values obtained from thermographic measurements on monocrystalline modules.

Cell	*T_r-max_* (°C)	*T_r-min_* (°C)	*U*(*T_r_*) (K)
E3	44.2	22.2	9.5
C4	44.8	23.6	9.5
B6	49.7	34.3	9.5

**Table 5 sensors-22-05685-t005:** Temperature values obtained from measurements by contact on the polycrystalline modules.

Cell	*T_c-max_* (°C)	*T_c-min_* (°C)	*U*(*T_c_*) (K)
E3	44.1	35.4	5.6
C4	42.1	30.7	5.8
B6	45.7	37.7	6.4

**Table 6 sensors-22-05685-t006:** Temperature values obtained from thermographic measurements on the polycrystalline modules.

Cell	*T_r-max_* (°C)	*T_r-min_* (°C)	*U*(*T_r_*) (K)
E3	36.6	29.4	9.5
C4	33.6	25.2	9.5
B6	39.5	33.0	9.5

## Data Availability

The data presented in this study are available on request from the corresponding author.

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
