# Peer review of "Evaluation of the Uncertainty of Surface Temperature Measurements in Photovoltaic Modules in Outdoor Operation"

_sensors, 2022, doi:10.3390/s22155685_

Round 1

Reviewer 1 Report

Please see uploaded review report.

Reviewer 2 Report

This paper presents statistical evaluations of the uncertainty of infrared (IR) and back-panel contact measurements in a monocrystalline module and a polycrystalline module.

Although the paper is attractive in its methodology and easy to understand despite the multiple layout errors, in this reviewer's opinion, the research and the manuscript are not yet in a mature form to be published.

Layout concerns

The paper needs a full orthographic and grammar verification; preferably, ask a native speaker or professional service to check the entire manuscript.

On the other hand, please verify the following: 

- 39-40 (does not seem like a correct justification).

- 122-124 (Figure does not show that).

- 158-159 (affirmation must be based).

- 173 Divisor 2? (please be clear).

- Some terminology and acronyms used in figures are missing; these must also match the text.

- Subsections must not have the same name.

- In table 2, the description is in Spanish.

Content/background concerns

Research on internal temperature estimation for PV panels is extensive, including IR systems with cheaper sensors such as the MLX90614; in this reviewer's opinion, the state-of-the-art verification needs to be completed.

Since a thermocouple is installed on the back side of the solar cells, the temperature is quite different from the front side, as the authors confirm at the end of the paper. Authors surely know that panels are composed of several layers that (could) act as temperature isolation layers, depending on manufacturer techniques. How do the authors cope with this problem or justify taking this temperature and generalizing it for EVERY PV module? Which is the objective of comparing these back-panel measurements (imprecise) with IR measurements during a specific period and environmental conditions such as moisture, dust, and surface degradation?

In line 206, in [19], the mentioned error cannot be taken for your scenario or must clearly state how. Indeed, Figure 7(a)-7(b) shows an 18% error approximately, and the Delta T data in Figure 10 is inconsistent for all cells; if one introduces a different brand PV module, the uncertainty could be higher.

Even more, moisture, dust, surface degradation, and other phenomena are inevitable and directly affect an IR measurement system, and these are not considered in the study.

Additionally, in this reviewer's opinion, realistic control data must be used to determine uncertainty levels.

Round 2

Reviewer 1 Report

The authors have adequately addressed the comments of the reviewer. 

The author can refer to the comments below to improve the quality of the paper.

1.       It is recommended to add to the text regarding the answer to question 3. The reason for using the two types of modules is not only the difference in materials but also the thermal characteristics of the modules. I think this explanation is important.

2.       In an outdoor environment, the reflected solar radiation is incident on the back surface. Therefore, the emissivity of the tape affects the temperature measurement accuracy of the thermocouple. Thermocouples show relatively high temperatures when black tape is used, and relatively low temperatures when metal tape is used. However, this effect may be negligible if the amount of solar radiation incident on the back surface of the module is small.

3.       The operating state of the PV module affects the temperature of the PV module. Therefore, it is recommended to add an explanation that the module operated at the maximum power point in the shaded state. My understanding is that the maximum power point is not the catalog spec, but the maximum power point in the I-V curve.

Reviewer 2 Report

Although the authors considered most of the concerns of this reviewer, key comments were no correctly aimed at the second version of the paper; specifically: "Research on internal temperature estimation for PV panels is extensive, (...) in this reviewer's opinion, the state-of-the-art verification needs to be completed."

Concerning the response: "Indeed, the expanded uncertainty in the thermocouple measurement, in percentage value, is around 20%. The value (...) Delta T gives an indication of the temperature difference between the two sides of the panel, (...)" but not with the actual cell temperature. How is such uncertainty estimation methodology enough to determine hot-spots, with that Delta T levels (not in the real PV surface and a 10 °C error).

The last response does not respond to the concern, and it is unethical to superimpose supposed credentials in place of scientific knowledge. The question is quite simple: why not realistic control data (actual PV panel temperature) was used (instead of back panel temperature)?

Additional comments:

- Line 79, what/which conductor?

- The authors may wish to clarify the usefulness of their study for future research or engineering applications. Perhaps the authors want to explain the usefulness of their analysis for future research or engineering applications since the measurement of PV temperature by infrared means and the estimation of its precision in this and other variables are not new and have been presented in various investigations. Note that no new references/state-of-the-art were considered (and/or marked), as suggested by this reviewer.

- Line 139 makes not much sense; complete the phrase.

- Check your title capitalization.

- Lines 464-469, the authors use "radiation measurement", which is incorrect.

Round 3

Reviewer 2 Report

None additional